# Measuring three-flavor neutrinos with FASER$\nu$ at the LHC

**Tomoko Ariga[1]$^\star$, on behalf of the FASER collaboration**

**1** Kyushu University

$\star$ tomoko.ariga@cern.ch

## Abstract

**FASER$\nu$ at the Large Hadron Collider (LHC) is designed to directly detect neutrinos from a collider for the first time and study their cross sections at TeV energies. To date, no such measurements have been made. In 2018, a pilot detector was installed and collected proton-proton collision data of 12.2 fb$^{-1}$ at a center-of-mass energy of 13 TeV. We observed the first neutrino interaction candidates at the LHC, opening a new avenue for studying neutrinos from current and future high-energy colliders. During Run 3 of the LHC from 2022, we will deploy a neutrino detector with a target mass of 1.1 tons. We expect to collect $\sim$2,000 $\nu_e$, 6,000 $\nu_\mu$, and 40 $\nu_\tau$ interactions in the detector.**

## 1 Introduction

There has been a longstanding interest in studying neutrinos produced at the Large Hadron Collider (LHC), although collider neutrinos have not been directly detected. In Run 3 of the LHC, proton–proton collisions at a center-of-mass energy of 14 TeV and with an expected integrated luminosity of 150 fb$^{-1}$ will produce a high-intensity beam of $\mathcal{O}(10^{12})$ neutrinos in the far-forward direction with a mean interaction energy of $\sim$1 TeV. In the ForwArd Search ExpeRiment (FASER), FASER$\nu$ [1] was designed to detect these neutrinos and study their properties. The FASER$\nu$ proposal [2] was approved in December 2019. The detector is being installed 480-m downstream of the ATLAS interaction point in the unused tunnel TI12. Beam exposure and data collection will be conducted from 2022. FASER$\nu$ is deployed on the beam collision axis to maximize the interaction rate of neutrinos of all three flavors, namely $\nu_e$, $\nu_\mu$, and $\nu_\tau$. This allows FASER$\nu$ to measure the interaction cross sections in the TeV energy range, which is currently unexplored. Figure 1 shows the constraints on neutrino charged-current interaction cross sections [3–7] and the expected energy spectra of the neutrinos interacting in FASER$\nu$. The FASER$\nu$ measurements can probe the gap between accelerator measurements ($E_\nu < 360$ GeV) [5] and the IceCube data ($E_\nu > 6.3$ TeV) [6–8] for muon neutrinos. For electron and tau neutrinos, cross section measurements can be extended to considerably higher

energies. In addition to the measurements of charged-current interactions, neutral-current interactions can be measured. Such measurements can provide a new limit on nonstandard interactions of neutrinos to complement the limits known from other measurements [9].

Furthermore, forward particle production, which is poorly constrained by the other LHC experiments, can be studied using FASER$\nu$. In particular, FASER$\nu$ measurements of electron neutrinos at high energies above $\sim$500 GeV, which mainly originate from charm decays [10], can provide the first data on forward charm production. In the case of neutrino telescopes such as IceCube, accelerator measurements of high-energy and large-rapidity charm production are necessary for the precise measurement of the cosmic neutrino flux. As a 14-TeV proton–proton collision corresponds to a 100-PeV proton interaction in the fixed-target mode, a direct measurement using FASER$\nu$ can provide important data for current and future high-energy neutrino telescopes.

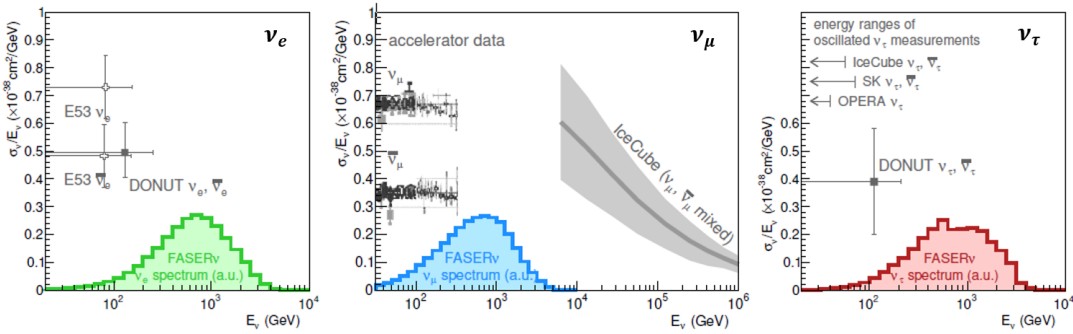

Figure 1: Constraints on neutrino charged-current interaction cross sections [3–7], and the expected energy spectra of neutrinos interacting in FASER$\nu$ [1].

## 2 The FASER experiment

FASER [11] is an experiment at the LHC with the goal of searching for light, weakly interacting particles such as dark photons and axion-like particles. Other searches for new physics at the LHC focus on high $p_T$ (appropriate for heavy, strongly interacting particles). To complement such searches, a particle detector will be located 480-m downstream of the ATLAS interaction point along the beam collision axis. The letter of intent [12] and technical proposal [13] for this experiment were reviewed, and approved by CERN in March 2019.

In addition to searches for new particles, we also proposed to study high-energy neutrinos of all flavors since the FASER location is also ideal for providing the first detection and studies of high-energy neutrinos produced at the LHC. The neutrinos at the LHC originate from the decays of forward-going hadrons, in particular, pions, kaons, hyperons, and charmed hadrons. A high-intensity beam of neutrinos is then produced in the far-forward direction. The expected numbers of neutrino interactions in FASER$\nu$ during LHC Run 3 are 2,000 $\nu_e$, 7,000 $\nu_\mu$, and 50 $\nu_\tau$. The differences between the generators are verified and summarized in [10].

## 3 First neutrino interaction candidates at the LHC

In 2018 during LHC Run 2, we performed a pilot run in the TI18 tunnel of the LHC tunnel to demonstrate neutrino detection at the LHC for the first time. We collected 12.2 fb$^{-1}$ during four weeks of data taking with pp collisions at 13-TeV center-of-mass energy. The pilot detector could not identify muons because its depth was only $0.6\lambda_{\text{int}}$, which is much shorter than

the $8\lambda_{\text{int}}$ of the full FASER$\nu$ detector which is prepared for LHC Run 3. The data from the pilot detector are used to prove the feasibility of high-energy neutrino measurements in this experimental environment.

The pilot detector consists of a module with 101 1-mm-thick lead plates and a module with 120 0.5-mm-thick tungsten plates, each containing the corresponding number of emulsion films. We searched for neutrino interactions by analyzing the data corresponding to 11 kg of the target mass and 18 neutral vertices passed the vertex selection criteria. We applied a multivariate discriminant based on a BDT algorithm to distinguish neutrino signal from neutral hadron background in the neutral vertex sample. We observed the first candidate events to be consistent with neutrino interactions at the LHC [14]. Figure 2 shows two candidate events. A 2.7$\sigma$ excess of neutrino-like signals over muon-induced backgrounds was measured. These results demonstrate the ability of FASER$\nu$ to detect neutrinos at the LHC and thus pave the way for future collider neutrino experiments.

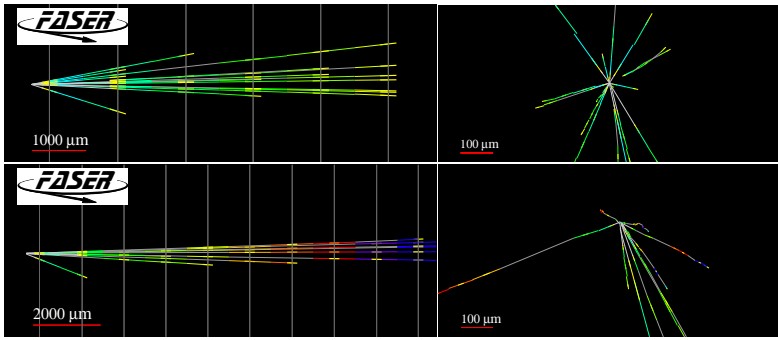

Figure 2: Event displays of two neutral vertices [14] in the $y$–$z$ projection longitudinal to the beam direction (left) and in the view transverse to the beam direction (right).

# 4 Detector for LHC Run 3

The FASER$\nu$ detector is located in front of the main FASER detector [13] along the beam collision axis. Figure 3 (top) shows the FASER$\nu$ detector and the main FASER detector. The FASER$\nu$ detector includes a veto station, an emulsion/tungsten detector, and an interface tracker (IFT) coupled to the FASER magnetic spectrometer. The emulsion/tungsten detector is designed to identify different lepton flavors which will be produced in $\nu_e$, $\nu_\mu$, $\nu_\tau$ interactions. It has finely sampled detection layers to identify electrons and to distinguish them from gamma rays, sufficient target material to identify muons, and good position and angular resolutions to detect tau and charm decays. Additionally, the detector can measure muon and hadron momenta and energy of electromagnetic showers, which will be used to estimate energy of neutrinos. The IFT is located downstream of the emulsion/tungsten detector to make the global analysis using both the emulsion/tungsten detector and the FASER spectrometer possible. Figure 3 (bottom) shows the topology of a neutrino event signal to be seen in the IFT and the FASER spectrometer.

The emulsion/tungsten detector consists of a repeated structure of emulsion films interleaved with 1-mm-thick tungsten plates. The emulsion film has two emulsion layers, each 70-$\mu$m thick. These layers are added onto both sides of a 210-$\mu$m-thick polystyrene base. The emulsion detector contains a total of 770 emulsion films, each of dimensions 25 cm $\times$ 30 cm. The total tungsten mass is 1.1 tons.

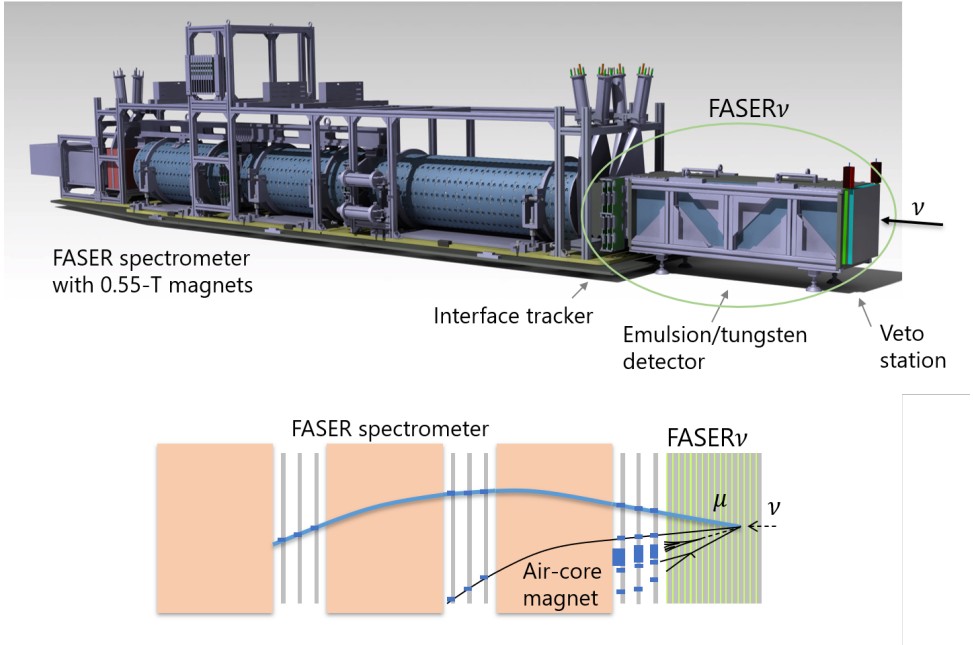

Figure 3: Sketch of the FASER detector (top) and the topology to be seen in the interface tracker and the FASER spectrometer (bottom).

In March 2021, the main FASER detector was successfully installed into the TI12 tunnel. Figure 4 shows a picture of the installed detector. The FASER$\nu$ veto station and the IFT were assembled and were used in a beam test at the H2 beamline of the CERN Super Proton Synchrotron. These detectors were installed in the tunnel in November 2021. The emulsion/tungsten detector will be installed just before the beam starts and will be replaced during planned technical stops of the LHC. The production of emulsion gel and films is scheduled few months before each replacement.

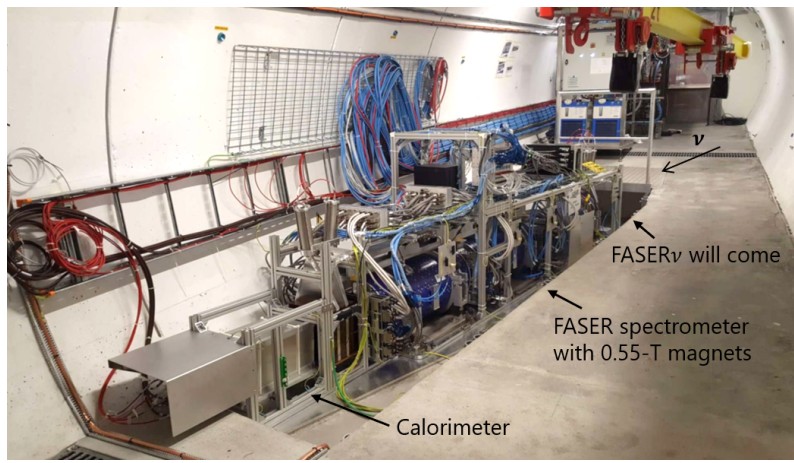

Figure 4: Picture of the main FASER detector installed in the TI12 tunnel.

## 5 Prospects for the high-luminosity LHC era

Toward the high-luminosity LHC era, we are planning to conduct the FASER$\nu$2 experiment at a proposed facility, the Forward Physics Facility (FPF). It is proposed to house a suite of experiments and to greatly enhance the physics potential of the LHC for searches related to physics beyond the standard model, neutrino physics, and QCD. The current status of plans for the FPF is summarized in [15]. FASER$\nu$2 is designed to perform precision measurements of high-energy neutrinos, especially, for tau neutrinos, and heavy-flavor physics studies. The FASER$\nu$2 detector will be a much larger successor to FASER$\nu$. FASER$\nu$2 is currently envisioned to be an emulsion-based detector with the target mass of 20 tons and to include a veto detector and interface detectors coupled to the FASER2 spectrometer. The high muon background in the LHC tunnel might be the experimental limitation for FASER$\nu$2. The possibility of sweeping away such muons using a magnetic field placed about few 100-m upstream of the detector is currently being explored.

## 6 Conclusion

FASER$\nu$ is designed to detect collider neutrinos of all three flavors for the first time and provide new measurements of their cross sections at energies higher than those detected from any previous artificial sources. It can facilitate the search for new effects in high-energy neutrino interactions. In the data acquired during LHC Run 2 in 2018, we observed the first neutrino interaction candidates at the LHC. Currently, the preparation of the FASER$\nu$ detector for obtaining data in LHC Run 3 is underway. With a deeper detector and lepton identification capability, FASER$\nu$ will perform better than the 2018 pilot detector. In 2022–2024, during LHC Run 3, we expect to collect ∼10,000 flavor-tagged charged-current neutrino interactions, along with to neutral-current interactions.

## Acknowledgments

We acknowledge CERN for the excellent performance of the LHC and the technical and administrative staff members at all FASER institutions. We gratefully acknowledge invaluable assistance from many groups at CERN, particularly, the Physics Beyond Colliders study group, the ATLAS Collaboration for providing the luminosity value, and the NA65/DsTau Collaboration for providing spare emulsion films for the 2018 run.

**Funding information** This work was supported partly by Heising-Simons Foundation Grant Nos. 2018-1135, 2019-1179, and 2020-1840, Simons Foundation Grant No. 623683. This work was supported by JSPS KAKENHI Grant Nos. JP19H01909, JP20H01919, JP20K04004, JP20K23373, a research grant from the Mitsubishi Foundation, and the joint research program of the Institute of Materials and Systems for Sustainability.

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
