# Peer review of "Measuring three-flavor neutrinos with FASERnu at the LHC"

_SciPost Physics Proceedings, doi:SciPost Phys. Proc. 16, 028 (2025)_

## Round 1 · Referee Report · Anonymous (Referee 1) · 2022-2-17

Strengths

1 - Clarity and conciseness
2 - First neutrino events at pilot-detector
3 - Implications for the future

Report

I think this proceeding is well written and well suited for the proceeding issue "16th International Workshop on Tau Lepton Physics (TAU2021)". I would recommend publication after minor revisions below.

Requested changes

1 - "Figure 1 shows the existing constraints...". There are additional measurements with IceCube using all-flavor data. Suggest removing "the existing" and giving references to those cross section measurements (https://doi.org/10.1103/PhysRevD.104.022001, https://doi.org/10.1103/PhysRevLett.122.041101).

2 - In the last paragraph of Sec. 1 it is stated that nue mainly originate from charm decays. This is energy dependent I think, as lighter mesons dominate at lower energies. I suppose it's true for FASERnu. It might be helpful to have a breakdown of the fluxes included as a figure or in the text.

3 - Fig. 1 right panel has SK extending to higher energies than IC-DeepCore. Could these numbers be double checked? In looking at the paper SK expectations drops below 1 at around 40 GeV, where as in IC there seems to be data at that energy still.

  1. Sec. 3 could use a bit more detail on the pilot-detector setup.

5 - Sec. 4 "detector is consisted" --> "detector consists"

6 - Sec. 4 "will be installed in the tunnel in November 2021." The proceeding is dated Nov. 24 2021 so I think the tense should be changed?

---

## Round 2 · Author Response

Dear Editor and Referee,
Thank you for reviewing our manuscript carefully and providing valuable comments. Please find below our response to your comments.
- We removed "the existing", and added references [7][8] in the text.
- We added reference[10] for such a figure and rephrased the text "electron neutrinos at high energies above $\sim$500 GeV, which mainly originate from charm decays".
- According to SK (http://arxiv.org/1711.09436) and IceCube (https://arxiv.org/abs/1901.05366), the tau neutrino appearance analysis is done with data up to 70 GeV (SK) and 56 GeV (IceCube). The plot indicates these energy ranges.
- We added a bit more details.
- We corrected.
- We corrected.

---

## Round 2 · List of Changes

As Listed above, we changed the following points.
- We added references [3][4][7][8].
- In the last paragraph of Sec. 1, we rephrased the text "electron neutrinos at high energies above $\sim$500 GeV, which mainly originate from charm decays".
- We added a bit more detail in Sec. 3.
- We corrected "detector is consisted" -> "detector consists" in Sec. 4.
- We corrected "will be installed in the tunnel in November 2021" -> "were installed in the tunnel in November 2021" in Sec. 4.

---

## Editorial Decision

published